# Temporal dynamics of pro-inflammatory cytokines and serum corticosterone following acute sleep fragmentation in male mice

**Van Thuan Nguyen, Cameron J. Fields, Noah T. Ashley** *

Department of Biology, Western Kentucky University, Bowling Green, Kentucky, United States of America

* noah.ashley@wku.edu

## Abstract

Obstructive sleep apnea is increasing worldwide, leading to disordered sleep patterns and inflammatory responses in brain and peripheral tissues that predispose individuals to chronic disease. Pro-inflammatory cytokines activate the inflammatory response and are normally regulated by glucocorticoids secreted from adrenal glands. However, the temporal dynamics of inflammatory responses and hypothalamic-pituitary-adrenal (HPA) axis activation in relation to acute sleep fragmentation (ASF) are undescribed. Male C57BL/6J mice were exposed to ASF or control conditions (no ASF) over specified intervals (1, 2, 6, or 24 h) and cytokine gene expression (IL-1β, TNF-α) in brain and peripheral tissues as well as serum glucocorticoid and interleukin-6 (IL-6) concentration were assessed. The HPA axis was rapidly activated, leading to elevated serum corticosterone from 1–24 h of ASF compared with controls. This activation was followed by elevated serum IL-6 concentration from 6–24 h of ASF. The tissue to first exhibit increased pro-inflammatory gene expression from ASF was heart (1 h of ASF). In contrast, pro-inflammatory gene expression was suppressed in hypothalamus from 1 h of ASF, but elevated at 6 h. Because the HPA axis was activated throughout ASF, this suggests that brain, but not peripheral, pro-inflammatory responses were rapidly inhibited by glucocorticoid immunosuppression.

## Introduction

The increased prevalence of obesity in the United States and other developed countries has drastically increased obstructive sleep apnea diagnoses [1]. This condition leads to sleep fragmentation (SF), reduced blood oxygen saturation, increased daytime sleepiness, and the occurrence of inflammation in the brain and peripheral tissues [2,3]. While obstructive sleep apnea is related to the development of chronic pathologies, such as metabolic [4,5] and cardiovascular diseases [6,7] as well as neurological disorders [8], the underlying mechanism of these studies is still unclear, although chronic inflammation is thought to play a large role in determining disease outcomes [2].

Inflammation is a pervasive phenomenon that is typically triggered during the onset of infection, injury, or exposure to pollutants [9]. Sleep loss is also a potent inducer of

**Data Availability Statement:** All datasets are available from Dryad (https://doi.org/10.5061/dryad.tdz08kq3p).

**Funding:** This research was supported by the following National Institutes of General Medical Sciences grants: R15GM117534-02 to NTA and P20GM103436-22 to KY-INBRE. https://www.nigms.nih.gov/. The funders had no role in study design, data collection and analysis, decision to publish, or preparation of the manuscript.

**Competing interests:** The authors have declared that no competing interests exist.

inflammation [2,3,10], but the mechanisms underlying this response are unclear. Sleep promotes the clearance of metabolic waste products, such as beta-amyloid protein [11–14], and loss of sleep, in turn, reduces clearance, leading to a build-up of waste products that may trigger the immune system to produce inflammatory mediators, termed herein the "metabolic clearance hypothesis." Based upon this possibility, the onset of inflammation from sleep loss could occur in the brain. The alternative hypothesis is that inflammation begins in the periphery with sympathetic afferents relaying this information to the brain, similar to a neuro-immune reflex to peripheral immune challenge [15]. As evidence, previous studies have shown that inhibition of the peripheral sympathetic nervous system (SNS) reduces inflammatory responses to SF in peripheral tissues [16,17] and brain [18].

Superimposed upon these inflammatory responses is activation of the hypothalamic-pituitary-adrenal (HPA) axis, which culminates in the release of glucocorticoids from the adrenal cortices and acts as a brake on immune function and inflammation [19]. Despite this long standing dogma, glucocorticoids can in some cases prime pro-inflammatory responses in the brain [20,21]. Interestingly, HPA axis activation occurs in tandem with pro-inflammatory responses to SF in mice [10,16,22,23]. Thus, it remains unresolved whether HPA activation provides negative feedback to inflammatory responses, temporarily potentiates responses, or has no effect.

The aim of this study was to evaluate the time course of peripheral and brain inflammatory responses in relation to HPA activation in male C57BL/6J mice exposed to acute SF. We predicted an initial increase in pro-inflammatory cytokine gene expression in brain that would be negatively regulated by a rise in serum corticosterone from HPA activation occurring later in the time course. We also predicted that neuroinflammatory responses would occur earlier than peripheral inflammatory responses, as suggested by the metabolic clearance hypothesis, which would represent the initial neuro-immune response to the build-up of metabolic waste products in brain. The alternative hypotheses are that the inflammatory response to sleep fragmentation occurs first in the periphery and is then relayed secondarily to the brain through sympathetic afferents or that peripheral and brain inflammatory responses operate independently of each other.

## Materials and methods

### Animals

Male adult C57BL/6J mice between 8–12 weeks of age (mean body mass: 26.8 ± 0.8 g) were used in this study ($n$ = 110; Jackson Laboratory, Bar Harbor, ME). Female mice were not assessed in this study to control for sexual differences in inflammatory/immune responses [24]. However, we are currently conducting a separate study in female mice to investigate whether responses vary according to sex. Mice were given rodent chow (Rodent RM4 1800, Cincinnati Lab Supply Inc.) and tap water *ad libitum* and housed in standard polypropylene mouse cages with corn cob (Combo Bed-o-cobs, Cincinnati Lab Supply Inc.) and enrichment (Enrich-n' nest paper blend, Cincinnati Lab Supply Inc.) bedding (lights on: 0800–2000 h, Central Standard Time, 21˚C ± 1˚C) at Western Kentucky University. Acute sleep fragmentation (ASF) experiments were performed using automated sleep fragmentation chambers (Lafayette Instrument Company; Lafayette, IN; model 80390) with a thin layer of corn cob bedding as previously described and each chamber contained no more than five mice [23]. These chambers ensure that mice are subjected to sleep fragmentation and not absolute sleep deprivation [16]. Mice were acclimated to the sleep fragmentation (SF) chambers for 48 h before the commencement of experiments to minimize carryover effects from the different cage environments [25]. This study was conducted under the approval of the Institutional Animal Care and

Use Committee at Western Kentucky University (#19–11), and procedures followed the National Institutes of Health's "Guide for the Use and Care of Laboratory Animals" and ARRIVE guidelines.

## Acute sleep fragmentation (ASF) and sample collection

Starting at 0800 (lights on; *Zeitgeber* Time (ZT)0), mice were exposed to 1 (ZT1), 2 (ZT2), 6 (ZT6), 12 (ZT12), or 24 (ZT24) h (n = 110; all groups, *n* = 10) of ASF, which involves a sweeping bar that moves horizontally across the modified cage every 120 sec, simulating the rate of SF in patients with severe sleep apnea [26] (Fig 1). For the no sleep fragmentation (control; CON) mice, subjects were housed in SF chambers, but no sweeping bar movements occurred. The CON groups matched collection times of ASF mice (all groups, *n* = 10). Both ASF and CON groups were compared to a baseline group of mice collected at 0800 CST (ZT0; *n* = 10).

After ASF or CON treatments, mice were rapidly anesthetized using isoflurane induction (5%) and decapitated <3 min of initial handling for tissue gene expression studies and blood collection for measurement of CORT and interleukin-6 (IL-6) levels (see below). Trunk blood was collected from mice, kept on ice for <20 min, and spun at 3000×g for 30 min at 4˚C. Serum was collected and stored at -80˚C for corticosterone and IL-6 ELISA assays (see below). For gene expression analyses, three brain regions (prefrontal cortex (PFC), hypothalamus, and hippocampus), and subsamples of liver, spleen, heart, and epididymal white adipose tissue (EWAT) were dissected from mice and stored in RNAlater solution (ThermoScientific) in the freezer at -20˚C. Dissection of brain regions followed Meyerhoff *et al.* [27]. These particular brain regions and peripheral tissues were chosen because previous studies have demonstrated elevated pro-inflammatory gene expression from ASF [18]. All tissue samples were stored at -20˚C before RNA extraction.

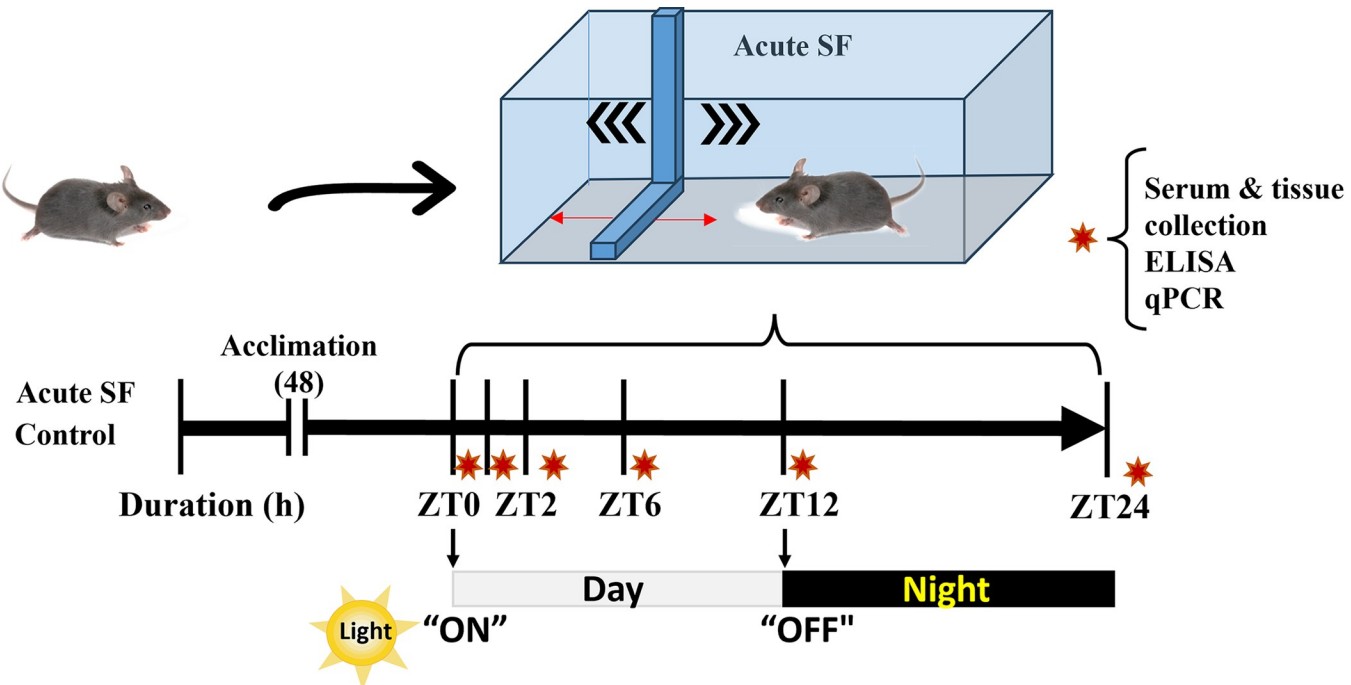

**Fig 1. Experimental protocol for acute sleep fragmentation (ASF) time-course study.** Mice were exposed to 1, 2, 6, 12, or 24 h of ASF, which involved a sweeping bar that moves horizontally across a modified cage every 120 s. Mice were acclimated to the sleep fragmentation chamber cage (no bar movement) for 48 h before starting the experiment. Control (CON; no sleep fragmentation) mice experienced no bar movement. ZT0 indicates when lights turned on; ZT12 indicates when lights turned off.

## Corticosterone and Interleukin-6 ELISA

Serum levels of corticosterone (n = 9-10/group) were measured using an ELISA kit (Catalogue number ADI-901-097, EnzoLife Sciences) which had a sensitivity of 26.99 pg/ml with cross-reactivity of <30% deoxycorticosterone and <2% progesterone. Samples were diluted 1:40 before running. The reaction was carried out in duplicate according to the kit instructions, and the average absorbance of the plate was determined using a plate reader (BioTek Synergy H1 Hybrid Reader). Average intra- and inter-assay variations were 2.85% and 2.65% respectively. IL-6 was measured in sera using ELISA MAX Deluxe kits (Catalogue number 431304; BioLegend, San Diego, CA). We decided to evaluate serum IL-6 because it has a lower detection limit in mouse sera compared with IL-1β or TNF-α (see ELISA MAX Deluxe kit specifications) and we only had enough serum from each mouse to measure one of these cytokines. The assays were carried out according to the manufacturer's instructions, and the average intra- and interassay variations were 8.24% and 7.43% respectively.

## Cytokine gene expression

RNA was extracted from liver, spleen, and epidydimal white adipose tissue (EWAT), as well as the prefrontal cortex, hippocampus, and hypothalamus from brain using a RNeasy mini kit (Qiagen). RNA was extracted from the heart using a RNeasy Fibrous Tissue mini kit (Qiagen). All extractions were performed following the manufacturer's instructions and were performed on <30 mg of tissue/sample. RNA concentrations were measured using a NanoDrop 2000 Spectrophotometer (Thermo Scientific). Total RNA was reverse transcribed using a high-capacity cDNA reverse transcription kit (ThermoFisher Scientific, Cat number: 4368813) according to the manufacturer's instructions and used as a template for determining relative cytokine gene expression using an ABI 7300 RTPCR system. Tissues were analyzed with cytokine primers/probes (IL-1β: Mm00434228, TNF-α: Mm00443258; ThermoFisher Scientific). Assay probes were labeled with florescent marker 5-FAM and quencher MGB at the 5' end and 3' end, respectively, and VIC-labeled 18S primer/probe (primer-limited; 4319413E; ThermoFisher Scientific) was used as an endogenous control. A multiplex PCR assay which included the genes of interest, and the endogenous control was run simultaneously for each sample. Samples were run in duplicate and the fold change in mRNA level was calculated as the relative mRNA expression levels, $2^{-\Delta\Delta Ct}$. The cycle threshold (Ct) at which the fluorescence exceeded background levels was used to calculate $\Delta Ct$ (Ct[target gene]–Ct[18S]). Each Ct value was normalized against the highest Ct value of a control sample ($\Delta\Delta Ct$), and then the negative value of this power to 2 ($2 -\Delta\Delta Ct$) was used for mRNA expression analysis.

## Statistical analysis

Data are presented as mean (±SE). All statistical analyses were performed using GraphPad Prism (version 9.0). Two-way ANOVAs assessed the effect of sleep treatment (ASF or CON), time (1h, 2h, 6h, 12h, 24h), and their interaction on mRNA expression of cytokines, serum CORT levels, and serum IL-6 concentration. One-way ANOVAs were used to assess whether ASF and CON groups differed significantly from baseline levels (time 0h). Tukey's HSD and Bonferroni multiple comparisons were used for post hoc analyses for one-way ANOVA and two-way ANOVA, respectively. $p < 0.05$ was considered statistically significant.

## Results

### Serum corticosterone concentration

Serum corticosterone levels were significantly affected by sleep treatment ($F_{1,86} = 131.2$, $p<0.0001$, Fig 2A), time ($F_{4,86} = 16.71$, $p<0.0001$) and their interaction ($F_{4,86} = 5.917$, $p =$

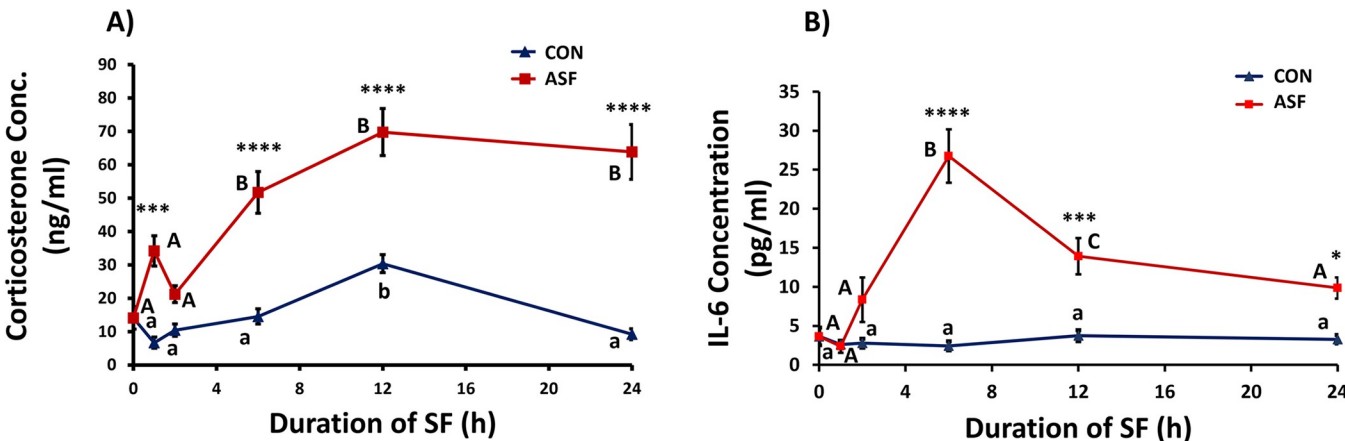

**Fig 2. Duration of acute sleep fragmentation (ASF) alters baseline glucocorticoid and IL-6 levels in serum.** A) Corticosterone (cort) concentration in male mice subjected to acute SF (0, 1, 2, 6, 12, and 24 h of ASF or no SF (CON)). Samples sizes are $n$ = 9–10 per group. B) IL-6 levels in male mice subjected to acute SF (0, 1, 2, 6, 12, and 24 h of ASF or no SF (CON)). Samples sizes are $N$ = 9-10/group. Significant effect of ASF (*** and **** denote p < 0.001 and 0.0001, respectively) relative to NSF at each time point was determined by two-way ANOVA followed by Bonferroni multiple comparisons post hoc tests. Differing lowercase and upper-case letters denote $p$ < 0.05 for NSF and CON groups, respectively and were analyzed using a one-way ANOVA and Tukey's HSD post hoc tests. Bar plots shown as means ± 1 SE and $p$ was set at 0.05 for statistical significance.

0.0003). ASF mice exhibited increased corticosterone levels compared with CON mice when sleep fragmentation lasted for 1, 6, 12, and 24h (Bonferroni post hoc test, $p$ = 0.0004, <0.0001, <0.0001, <0.0001, respectively). Among CON mice, serum corticosterone concentration ($F_{5,50}$ = 12.84, $p$<0.0001) increased from ZT0 to ZT12 and decreased from ZT12 to ZT24, respectively (ZT12 time point, Tukey's HSD post hoc, $p$ = 0.0002 and <0.0001 compared to other points). ASF mice exhibited increased serum corticosterone levels at ZT6 compared with baseline (ZT0; $F_{5,52}$ = 15.02, $p$<0.0001; Tukey's HSD post hoc, $p$ = 0.0005), and remained elevated above baseline at ZT12 ($p$ = <0.0001) and ZT24 ($p$ = <0.0001).

## Serum Interleukin-6 (IL-6) concentration

Sleep treatment, time, and their interaction significantly affected serum IL-6 levels ($F_{1,\,88}$ = 73.19, $p$<0.0001, $F_{4,\,88}$ = 13.73, $p$<0.0001, $F_{4,88}$ = 14.27, $p$<0.0001, respectively; Fig 2B). Specifically, ASF mice exhibited increased IL-6 levels compared with CON mice when sleep fragmentation lasted for 6, 12, or 24h (Bonferroni post hoc test, $p$<0.0001, $p$ = 0.0003, $p$<0.0371, respectively). CON mice displayed stable IL-6 levels ($F_{5,53}$ = 0.6383, $p$ = 0.6714) around 3–4 pg/ml while ASF mice had a rapid increase at ZT6 ($F_{5,53}$ = 16.24, $p$<0.0001; 6h, Tukey's HSD post hoc, $p$<0.0001), followed by a decrease at ZT12 and ZT24 (Tukey's HSD post hoc, $p$ = 0.0016, $p$<0.0001, respectively).

## Peripheral responses

**Spleen.** ASF had no effect on TNF-α ($F_{1,\,84}$ = 0.001368, $p$ = 0.9706) or IL1β ($F_{1,84}$ = 0.1895, $p$ = 0.6645) gene expression in spleen (Fig 3A and 3B). However, TNF-α ($F_{4,84}$ = 22.17, $p$ = 0.0012) and IL-1β expression ($F_{4,84}$ = 4.822, $p$ = 0.0015) were affected by time compared with CON mice, but there were no significant interaction effects (TNF-α: $F_{4,84}$ = 0.8545, $p$ = 0.4948; IL-1β: $F_{4,84}$ = 1.066, $p$ = 0.3785).

**Heart.** Sleep treatment had no overall effect upon TNF-α expression in heart ($F_{1,83}$ = 0.2545, $p$ = 0.6152; Fig 3C). There were significant effects of time ($F_{4,83}$ = 47.06, $p$<0.0001) and the interaction between sleep treatment and time ($F_{4,83}$ = 5.024, $p$ = 0.0011). More specifically,

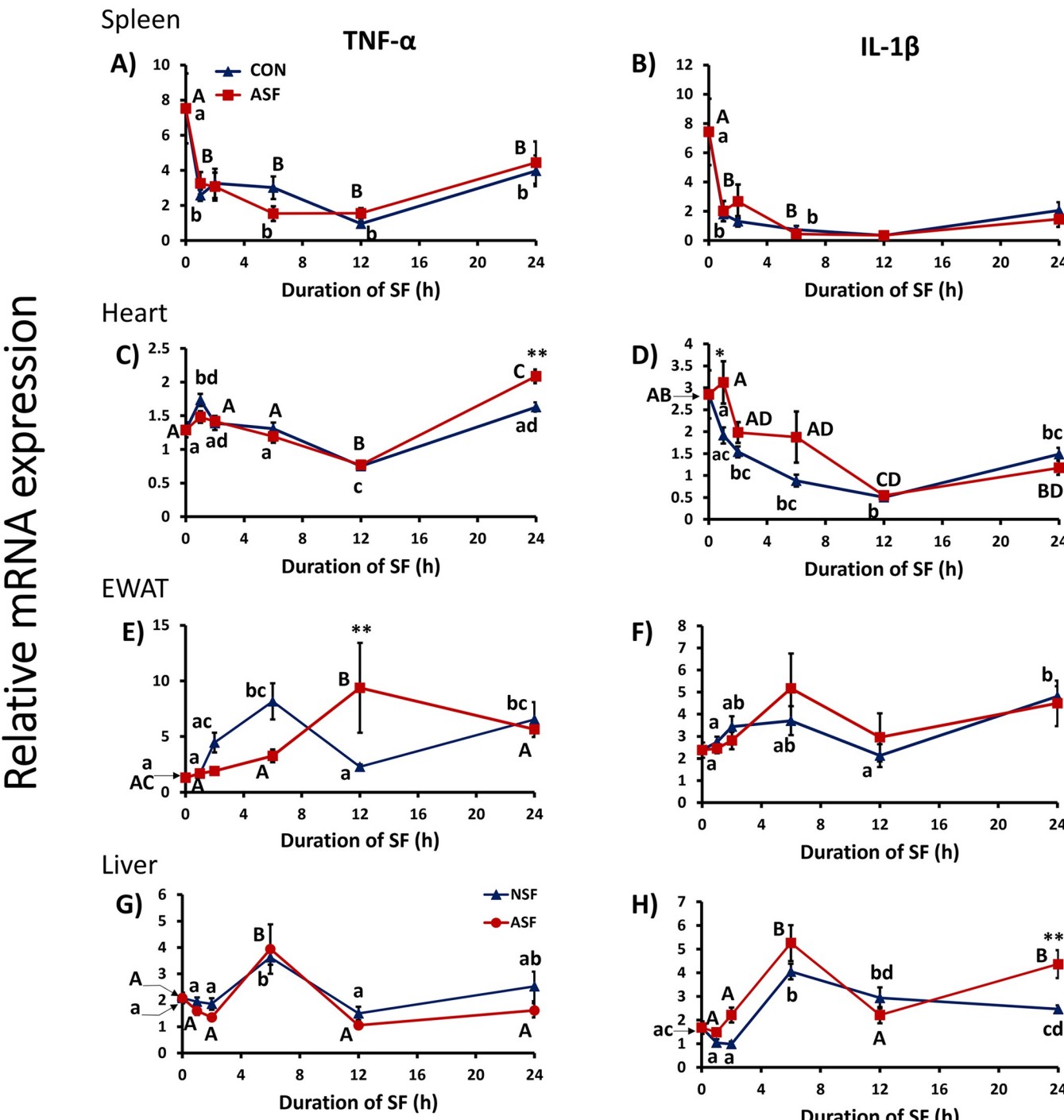

**Fig 3. Effects of sleep fragmentation, time, and their interaction on TNF-α and Il-1β gene expression in peripheral tissues.** Panels show TNF-α and Il-1β gene expression in spleen (A, B), heart (C, D), epididymal white adipose tissue (EWAT; E, F) and liver (G, H), respectively ($n$ = 8-10/group, time-course: 0, 1, 2, 6, 12, 24h). Significant effect of sleep treatment (** denotes $p < 0.01$) at each time point was determined by two-way ANOVA followed by Bonferroni multiple comparisons post hoc tests. Differing lowercase and upper-case letters denote $p < 0.05$ across different time points for CON and ASF, respectively and were analyzed using a one-way ANOVA and Tukey's HSD post hoc tests. Bar plots shown as means ± 1 SE and $p$ was set at 0.05 for statistical significance.

at ZT24, ASF mice exhibited increased TNF-α expression compared with CON mice (Bonferroni post hoc test, $p = 0.0014$). In contrast, there was a significant effect of sleep treatment ($F_{1,86} = 7.045$, $p = 0.0095$, Fig 3D), time ($F_{4,86} = 13.66$, $p < 0.0001$), and their interaction ($F_{4,86} = 2.509$, $p = 0.0477$) upon IL-1β expression in heart. At ZT1, ASF mice increased cardiac IL-1β expression relative to CON mice (Bonferroni post hoc test, $p = 0.0136$).

**EWAT.** Sleep treatment had no overall effect on TNF-α ($F_{1, 83} = 0.06913$, $p = 0.7933$) or IL-1β ($F_{1, 82} = 0.1820$, $p = 0.6708$) gene expression in EWAT (Fig 3E and 3F). However, time affected both TNF-α ($F_{4, 83} = 3.492$, $p = 0.0110$) and IL-1β ($F_{4, 82} = 3.340$, $p = 0.0138$), but there was only a significant interaction effect for TNF-α gene expression (TNF-α: $F_{4, 83} = 4.754$, $p = 0.0017$; IL-1β: $F_{4, 82} = 0.6524$, $p = 0.6268$). At ZT12, ASF mice has significantly higher TNF-α expression in EWAT compared with CON mice (Bonferroni post hoc test, $p = 0.0047$).

**Liver.** There was neither a significant effect of sleep treatment nor an interaction effect on hepatic TNF-α expression (sleep treatment, $F_{1, 82} = 2.254$, $p = 0.1371$; interaction: $F_{4, 82} = 0.5843$, $p = 0.6749$), but there was a significant effect of time on TNF-α expression in liver ($F_{4, 82} = 11.67$, $p < 0.0001$; Fig 3G). Sleep treatment, time, and their interaction had a significant effect on IL-1β gene expression in liver (sleep treatment: $F_{1, 81} = 10.81$, $p = 0.0015$; time: $F_{4, 81} = 25.37$, $p < 0.0001$; interaction: $F_{4, 81} = 3.332$, $p = 0.0141$). More specifically, IL-1β gene expression was significantly higher in ASF mice compared with CON mice at ZT24 (Bonferroni post hoc test, $p < 0.05$; Fig 3H).

## Brain responses

**Hippocampus.** Although sleep treatment or time course did not affect hippocampal TNF-α (Acute SF: $F_{1, 83} = 0.6730$, $p = 0.4144$; time course: $F_{4, 83} = 2.327$, $p = 0.0629$) gene expression in the hippocampus, the interaction effect was significant ($F_{4, 83} = 8.365$, $p < 0.0001$). ASF mice exhibited increased TNF-α expression in hippocampus compared with CON at ZT12 (Bonferroni post hoc test, $p = 0.0389$) and ZT24 (Bonferroni post hoc test, $p < 0.0003$) time points (Fig 4A). Additionally, sleep treatment, time, and their interaction had a significant effect on hippocampal IL-1β expression (sleep treatment: $F_{1, 88} = 16.94$, $p < 0.0001$; time: $F_{4, 88} = 6.622$, $p = 0.0001$; interaction: $F_{4, 88} = 6.650$, $p = 0.0001$; Fig 4B).

**Prefrontal cortex (PFC).** Sleep treatment had no effect on TNF-α or IL-1β (TNF-α: $F_{1, 86} = 0.08523$, $p = 0.7710$; IL-1β: $F_{1, 85} = 0.1395$, $p = 0.7097$). However, time had a significant effect on TNF-α and IL-1β (TNFα: $F_{4, 86} = 5.465$, $p = 0.0006$; IL1β: $F_{4, 85} = 30.24$, $p < 0.0001$) gene expression (Fig 4C and 4D), but there were no significant interaction effects.

**Hypothalamus.** Sleep treatment had no overall effect upon hypothalamic TNF-α expression ($F_{1, 85} = 0.07240$, $p = 0.7885$). However, there were significant effects of time ($F_{4, 85} = 2.535$, $p = 0.0460$) and the interaction between sleep treatment and time ($F_{4, 85} = 6.042$, $p = 0.0003$; Fig 4E). In hypothalamus, there were significant effects of sleep treatment, time, and their interaction on IL-1β gene expression (sleep treatment: $F_{4, 84} = 12.21$, $p = 0.0008$; time: $F_{4, 84} = 3.118$, $p = 0.0192$; interaction: $F_{4, 84} = 3.173$, $p = 0.0177$). More specifically, ASF mice exhibited significantly lower TNF-α gene expression level at ZT1 (Bonferroni post hoc test, $p = 0.0115$) and significantly higher levels gene expression at ZT12 (Bonferroni post hoc test, $p = 0.0289$, Fig 4E) and significantly higher IL-1β gene expression levels at ZT6 and ZT12 compared with CON mice (Bonferroni post hoc test, $p = 0.0103$, $p = 0.0165$, respectively, Fig 4F).

## Discussion

Sleep loss induces an inflammatory response in various tissues of the body [10], but the temporal dynamics of inflammatory responses are unclear. Initial results from this time-course study indicate that elevated pro-inflammatory gene expression was first detected in heart tissue after

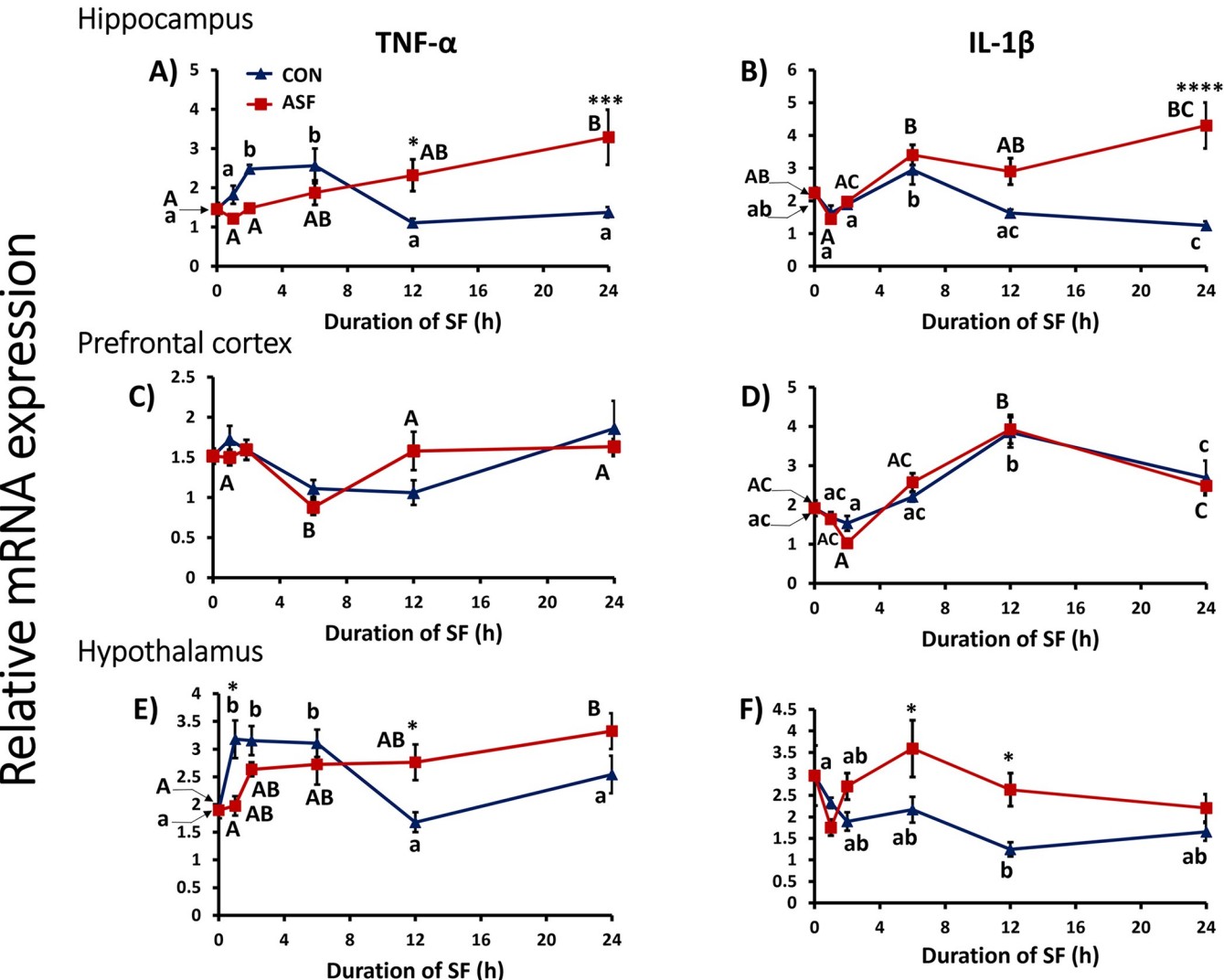

**Fig 4. Effects of sleep fragmentation, time, and their interaction on TNF-α and Il-1β gene expression in select brain regions.** Panels show TNF-α and Il-1β gene expression in HIP (A, B), PFC (C, D), and HYP (E, F), respectively (n = 9-10/group, Acute SF time-course: 0, 1, 2, 6, 12, 24h). Significant effect of SF (*, ** and *** denote p < 0.05, 0.01, and 0.001, respectively) was determined by two-way ANOVA followed by Bonferroni multiple comparisons post hoc test. Differing lowercase and upper-case letters denote p < 0.05 across different time points for CON and ASF, respectively and were analyzed using a one-way ANOVA and Tukey's HSD post hoc tests. Bar plots shown as means ± 1 SE and p was set at 0.05 for statistical significance. HIP: Hippocampus; PFC: prefrontal cortex; HYP: Hypothalamus.

1 h of sleep fragmentation. The first brain region measured that exhibited neuroinflammation was hypothalamus after 6 h of sleep fragmentation. However, there was suppression of pro-inflammatory gene expression in hypothalamus after 1 h of sleep fragmentation relative to controls. These findings provide some support for the alternative hypothesis that peripheral inflammation from sleep fragmentation occurs first in the periphery, specifically the heart, rather than brain. However, it should be emphasized that our first measurement following sleep fragmentation was 1 h; it is possible that an inflammatory response could have occurred even earlier (<60 min) from sleep fragmentation.

Previous studies have documented increased pro-inflammatory gene expression in cardiac tissue following short-term [10,16] and chronic sleep fragmentation [16,28]. These initial findings are consistent with a rapid elevation in pro-inflammatory gene expression in heart. We

surmise that pro-inflammatory cytokines are elevated rapidly in heart due to increased SNS activity from sleep fragmentation [23,29,30]. Whether the heart directly relays inflammatory "information" to brain through sympathetic afferents or whether neuroinflammation is simply a slower, but independent, process requires further study. In addition, because this study only measured select regions in brain and several organs/tissues in the periphery, it is possible that other tissues that were not measured may have different temporal responses to sleep fragmentation.

The HPA axis was rapidly activated from acute sleep fragmentation as measured by increased corticosterone concentration in serum, which is a consistent finding from previous studies [10,18,22,23]. Control mice exhibited a well-known diurnal rhythm in circulating corticosterone concentration with peak levels occurring at the onset of nocturnal activity [31]. Among mice subjected to sleep fragmentation, serum corticosterone increased rapidly from 0h to 1h and remained elevated for the entire time course compared with control mice. As stated above, at 1h of sleep fragmentation, there was suppression of TNF-α expression in hypothalamus, but an elevation of IL-1β expression in heart compared with controls. These findings imply that the increased serum corticosterone concentration at this 1h time point may be rapidly suppressing an inflammatory response in brain, such as hypothalamus, but has no suppressive effect or even possibly a pro-inflammatory effect in the periphery (Table 1).

Sleep fragmentation elevated serum IL-6 levels concomitant with increased serum corticosterone levels, with IL-6 levels peaking after 6 h but falling thereafter. Previous studies employing various forms of sleep deprivation on mice and humans have described elevated serum IL-6 concentration [32,33]. IL-6 acts pleiotropically through multiple pro- and anti-inflammatory pathways [34], and glucocorticoids can alter the balance of these pathways through interfering with the expression of the suppressor of cytokine signalling 3 (SOCS3) feedback inhibitor [35], as well as repressing the transcriptional activation of nuclear factor-kappa B (NF-κB) [36]. Whether elevated serum corticosterone concentration provided negative feedback to decrease serum IL-6 levels at 12 and 24 h after sleep fragmentation (relative to 6 h) will require additional investigation. In addition, it is also unknown whether IL-6 reciprocally activated the HPA axis as is often the case for bi-directional neuroendocrine-immune interactions [37].

Studies that manipulate glucocorticoid action either through adrenalectomy/hormone replacement experiments or pharmacological approaches that inhibit glucorticoid synthesis and/or receptor binding are needed to pinpoint the precise modulatory effects of glucocorticoids. It is also possible that activation of the SNS from acute sleep fragmentation through release of norepinephrine/epinephrine from adrenal medullae could promote a pro-inflammatory effect in the periphery that is independent of glucocorticoid effects. As evidence, suppression of the SNS using chemical sympathectomy alleviates inflammatory responses from acute and chronic sleep fragmentation in peripheral tissues [16] as well as brain [18]. Furthermore, it is conceivable that other hormones affected by sleep fragmentation (e.g., sex steroids) could also impinge upon inflammatory responses [24,38].

These findings indicate that pro-inflammatory responses to acute sleep fragmentation are tissue-specific, which is consistent with previous studies [16–18,23]. Acute sleep fragmentation increased TNF-α expression in heart and EWAT after exposure for 12 and 24 h, respectively, but there was no effect in liver or spleen. In addition, elevated TNF-α expression from sleep fragmentation occurred earliest in EWAT (for peripheral tissues) at 12h while others were elevated at 24h (Fig 3A, 3C and 3E and Table 1). This finding is consistent with previous studies, supporting the role of EWAT in the pro-inflammatory response to acute sleep fragmentation [22,23].

IL-1β expression was significantly elevated in hypothalamus after 6 and 12 h of sleep fragmention and in hippocampus after 24 h of sleep fragmentation (Fig 4 and Table 1). A similar

**Table 1. Summary of time course of pro-inflammatory cytokine gene expression among mice subjected to ASF vs. CON.**

| Modulation of pro-inflammatory gene expression (TNF-α, IL-1β) | | | | | |
|---|---|---|---|---|---|
| **Tissue** | **Acute SF time course compared with controls** | | | | |
| | **1h** | **2h** | **6h** | **12h** | **24h** |
| **Liver** | | | | | Up (IL-1β) |
| **Heart** | Up (IL-1β) | | | | Up (TNF-α) |
| **Spleen** | | | | | |
| **EWAT** | | | | Up (TNF-α) | |
| **Prefrontal cortex** | | | | | |
| **Hypothalamus** | Down (TNF-α) | | Up (IL-1β) | Up (IL-1β, TNF-α) | Up (TNF-α) |
| **Hippocampus** | | | | Up (TNF-α) | Up (IL-1β, TNF-α) |

Up- or downregulation of TNF-α and/or IL-1β gene expression (relative to controls) in relation to duration of SF (1, 2, 6, 12, or 24 h). Epididymal white adipose tissue: EWAT, SF: sleep fragmentation.

pattern for TNF-α gene expression was also observed occurring slightly later (Fig 4 and Table 1). Acute sleep fragmentation did not affect TNF-α or IL-1β expression in prefrontal cortex (Fig 4 and Table 1). These results are a departure from previous studies that have shown elevated pro-inflammatory gene expression in hypothalamus, hippocampus, and pre-frontal cortex after 24-h of sleep fragmentation among female C57BL/6J mice [17,18], but very few effects observed among males [10,23], although there was a non-significant trend for increased pro-inflammatory gene expression in hippocampus in one study[12]. Neuroinflammation is complex and involves contributions from neuron-glia interactions that may be playing a role in central inflammatory responses to sleep loss [39,40]. In this study, we only evaluated cytokine gene expression in manual microdissections of these brain regions. Future studies should evaluate the cellular basis of this neuroinflammatory response by assessing microglial and astrocyte morphology using immunocytochemistry, as well as utilizing spatial transcriptomics on brain slices to quantify transcripts of immune/inflammatory genes at distinct locations in these brain regions.

## Conclusions

Sleep fragmentation and other forms of perturbed sleep promote an inflammatory environment that predisposes individuals towards the development of chronic disease [2,3]. However, the mechanisms that lead to the onset of inflammation from sleep fragmentation are poorly understood. Does inflammation begin in the brain or the periphery? In this initial time-course study, we provide evidence that the heart is one of the first organs to produce elevated pro-inflammatory gene expression, which suggests that inflammation from sleep fragmentation is rapidly initiated in the periphery. Because glucocorticoids are also elevated during sleep fragmentation, our findings imply that glucocorticoids may rapidly suppress inflammatory responses in certain regions of the brain, like hypothalamus, but not in peripheral tissues, such as heart and white adipose tissue. Instead, the rapid activation of the sympathetic nervous system from sleep fragmentation is most likely promoting an inflammatory environment in peripheral tissues, while possibly overriding negative feedback from glucocorticoids.

It will be important to know whether these findings extend to our understanding of inflammatory profiles in response to chronic sleep fragmentation (e.g., obstructive sleep apnea, insomnia). Elucidating the temporal dynamics of inflammatory responses in relation to chronic sleep fragmentation (weeks to months) will represent critical next steps. However, identifying the physiological mechanisms that drive the onset of this inflammatory response from sleep fragmentation could provide promising avenues for the development of new therapeutic options for patients experiencing sleep dysfunction. If we can understand why, where, and when inflammatory responses occur from sleep loss, then therapeutics could be designed to mitigate these responses both temporally and spatially to improve patient outcomes.

## Acknowledgments

We thank Naomi Rowland for assistance with RT-PCR and the WKU Biotechnology Center for assistance and access to resources.

## Author Contributions

**Conceptualization:** Noah T. Ashley.

**Data curation:** Van Thuan Nguyen.

**Formal analysis:** Van Thuan Nguyen.

**Funding acquisition:** Noah T. Ashley.

**Investigation:** Van Thuan Nguyen, Cameron J. Fields, Noah T. Ashley.

**Methodology:** Van Thuan Nguyen, Noah T. Ashley.

**Project administration:** Noah T. Ashley.

**Supervision:** Noah T. Ashley.

**Visualization:** Van Thuan Nguyen.

**Writing – original draft:** Van Thuan Nguyen, Noah T. Ashley.

**Writing – review & editing:** Van Thuan Nguyen, Noah T. Ashley.

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
