## [Decision Letter · Decision Letter 0]

12 Oct 2023

PONE-D-23-20980Temporal Dynamics of Pro-inflammatory Responses and Glucocorticoid Release from Acute Sleep Fragmentation in Male MicePLOS ONE

Dear Dr. Ashley,

Thank you for submitting your manuscript to PLOS ONE. After careful consideration, we feel that it has merit but does not fully meet PLOS ONE’s publication criteria as it currently stands. Therefore, we invite you to submit a revised version of the manuscript that addresses the points raised during the review process.

ACADEMIC EDITOR: - please do follow directions from our reviewer to improve your study- could some of your results be explained by alteration in hormonal status affected by sleep fragmentation. Extend discussion to neuroinflammation and explain potential limitations and clinical benefits of your results more deeply.  ==============================

We look forward to receiving your revised manuscript.

Kind regards,

Prof. Dr. Dragan Hrncic, MD, PhD 

Academic Editor

PLOS ONE

2. We note that Figure 1 in your submission contain copyrighted images. All PLOS content is published under the Creative Commons Attribution License (CC BY 4.0), which means that the manuscript, images, and Supporting Information files will be freely available online, and any third party is permitted to access, download, copy, distribute, and use these materials in any way, even commercially, with proper attribution. For more information, see our copyright guidelines: http://journals.plos.org/plosone/s/licenses-and-copyright.

Reviewers' comments:

Reviewer's Responses to Questions

**Comments to the Author**

1. Is the manuscript technically sound, and do the data support the conclusions?

Reviewer #1: Yes

2. Has the statistical analysis been performed appropriately and rigorously? 

Reviewer #1: Yes

3. Have the authors made all data underlying the findings in their manuscript fully available?

Reviewer #1: Yes

4. Is the manuscript presented in an intelligible fashion and written in standard English?

Reviewer #1: Yes

5. Review Comments to the Author

Reviewer #1: This is a straightforward study that examines serum corticosterone and serum IL-6, as well as Il-1b mRNA and TNF-a mRNA) after experimental acute sleep fragmentation in male mice. The methods appear generally sound. However, I have some suggestions for improvement.

Major comments

- Title. I would suggest “Temporal dynamics of corticosterone and pro-inflammatory cytokines following acute sleep fragmentation in male mice” (or something similar).

- General comment. Fewer abbreviations would make the text easier to read (e.g. can remove “SF” and “OSA”). Why not use “control” or “CON” instead of “NSF”?

- Methods. Why were only male mice used? This should be explained.

- Methods. What was the body mass of the animals?

- Methods. Please provide more information on the water, diet, and bedding. This is an important part of the ARRIVE guidelines.

- Methods. For peripheral tissues, was the whole organ collected and used for analysis? How much tissue was collected for each brain region? How much tissue was used to measure gene expression?

- Methods. Why not measure serum IL-1b and TNF-a too? Why not measure IL-6 mRNA too? I did not understand the rationale for what was measured. This should be clearer.

- Results. Figures for relative mRNA expression: Differing lowercase and uppercase letters are difficult to differentiate between 0, 1 and 2 hours.

- Discussion. I am not sure *acute* sleep fragmentation reveals much about the effects of *chronic* apnea. This is worth a short paragraph in the Discussion.

Minor comments

- Figure 1. Consider indicating when the lights turn on (ZT0) and off (ZT12). Each rectangle during the duration of SF indicates 1 hour, but the two rectangles for the acclimation period represent two days (48 hours). Could clarify this in the legend.

- Figure 2B. The color and shape for NSF group (light blue circle) is different from the rest of the figures (dark blue triangle).

- Figure 4 legend. Typo: “PCF: prefrontal cortex”

- Results (line 270). Typo: “IL-1”

- Discussion (lines 347-348). This sentence is a bit confusing. Consider rewording the sentence.

6. PLOS authors have the option to publish the peer review history of their article (what does this mean?). If published, this will include your full peer review and any attached files.

Reviewer #1: No

---

## [Author Response · Author response to Decision Letter 0]

3 Nov 2023

We note that Figure 1 in your submission contain copyrighted images.

Thank you. We have made a new original figure to replace Figure 1.

ACADEMIC EDITOR:

- please do follow directions from our reviewer to improve your study

We believe that we have satisfactorily addressed the reviewer’s comments below. 

- could some of your results be explained by alteration in hormonal status affected by sleep fragmentation.

This raises an interesting point. It is possible that sleep fragmentation could alter other hormones (besides glucocorticoids) that could affect inflammatory responses, e.g., androgens. We discuss this possibility in the discussion.

 Extend discussion to neuroinflammation and explain potential limitations and clinical benefits of your results more deeply. 

We have provided more info on neuroinflammation and also discuss potential limitations and clinical benefits in the discussion.

Reviewer #1: This is a straightforward study that examines serum corticosterone and serum IL-6, as well as Il-1b mRNA and TNF-a mRNA) after experimental acute sleep fragmentation in male mice. The methods appear generally sound. However, I have some suggestions for improvement.

Major comments

- Title. I would suggest “Temporal dynamics of corticosterone and pro-inflammatory cytokines following acute sleep fragmentation in male mice” (or something similar).

Thank you. We have made this change and have added “serum” in front of corticosterone.

- General comment. Fewer abbreviations would make the text easier to read (e.g. can remove “SF” and “OSA”). Why not use “control” or “CON” instead of “NSF”?

Thank you. We have cut down on the abbreviations in the text as recommended. We have spelled out OSA. We also have replaced “NSF” with “CON” throughout texts and figures. However, we believe that the abbreviation for SF should remain because we discuss it so much in this study. In the discussion, we attempt to remove SF and replace it with “sleep fragmentation” to make it more palatable to readers.

- Methods. Why were only male mice used? This should be explained.

We only used male mice to control for variation in immune/inflammatory responses that are well known to occur between the sexes. Adult female mice undergo a 4-day estrus cycle that may complicate their inflammatory responses to sleep fragmentation. Nonetheless, we are currently evaluating the temporal responses of female mice to sleep fragmentation and hope to compare them to male mice at a later date.

- Methods. What was the body mass of the animals?

We now report the average body mass of male mice used in the paper. 

- Methods. Please provide more information on the water, diet, and bedding. This is an important part of the ARRIVE guidelines.

Thank you. We now report more information on the type of water used, the diet of the mice, and the type of bedding used. 

- Methods. For peripheral tissues, was the whole organ collected and used for analysis? How much tissue was collected for each brain region? How much tissue was used to measure gene expression? 

For peripheral tissues, a sub-sample of the whole organ was collected because the RNA extraction only permits less than 30 mg of tissue to be homogenized and measured for gene expression. Between 5-30 mg of brain tissue was collected for each brain region. 

- Methods. Why not measure serum IL-1b and TNF-a too? Why not measure IL-6 mRNA too? I did not understand the rationale for what was measured. This should be clearer.

Thanks for your comments. We were limited in the amount of serum from each mouse, so were not able to assess more than one serum cytokine. The sandwich ELISA that we used is most sensitive to IL-6, so we made the decision to assess this pro-inflammatory cytokine in the serum. We were worried that IL-1b or TNF-a in the serum could be non-detectable (from prior experience). We would have liked to have measured IL-6 mRNA as well, but most of the previous studies in our lab have focused mostly on IL-1-beta and TNF-alpha gene expression, and the Taqman reagents/primers/probes are so expensive, we could not afford to measure another gene for this study.

- Results. Figures for relative mRNA expression: Differing lowercase and uppercase letters are difficult to differentiate between 0, 1 and 2 hours.

Thank you. We see what you are referring to. We have attempted to move those around slightly to make them more discernible. 

- Discussion. I am not sure *acute* sleep fragmentation reveals much about the effects of *chronic* apnea. This is worth a short paragraph in the Discussion.

Thanks for pointing this out. We now provide a discussion on how the results are relevant to the etiology of chronic inflammatory diseases, like obstructive sleep apnea.

Minor comments

- Figure 1. Consider indicating when the lights turn on (ZT0) and off (ZT12). Each rectangle during the duration of SF indicates 1 hour, but the two rectangles for the acclimation period represent two days (48 hours). Could clarify this in the legend.

Thank you. We have now clarified this in the figure and the legend. 

- Figure 2B. The color and shape for NSF group (light blue circle) is different from the rest of the figures (dark blue triangle).

Thanks. We have now fixed this. 

- Figure 4 legend. Typo: “PCF: prefrontal cortex”

Thanks. This has been fixed

- Results (line 270). Typo: “IL-1”

Also fixed. 

- Discussion (lines 347-348). This sentence is a bit confusing. Consider rewording the sentence.

We have now reworded the sentence. “IL-1β expression was significantly elevated in hypothalamus after 6 and 12 h of sleep fragmentation and in hippocampus after 24 h of sleep fragmentation; Fig 4, Table 1). A similar pattern for TNF-α gene expression was also observed occurring slightly later (Fig 4, Table 1). Acute sleep fragmentation did not affect TNF-α or IL-1β expression in prefrontal cortex (Fig 4, Table 1).”

---

## [Decision Letter · Decision Letter 1]

29 Nov 2023

Temporal dynamics of pro-inflammatory cytokines and serum corticosterone following acute sleep fragmentation in male mice

PONE-D-23-20980R1

Dear Dr. Ashley,

We’re pleased to inform you that your manuscript has been judged scientifically suitable for publication and will be formally accepted for publication once it meets all outstanding technical requirements.

Kind regards,

Prof. Dr. Dragan Hrncic, MD, MSc, MBE, PhD

Academic Editor

PLOS ONE

Additional Editor Comments (optional):

Reviewers' comments:

Reviewer's Responses to Questions

**Comments to the Author**

1. If the authors have adequately addressed your comments raised in a previous round of review and you feel that this manuscript is now acceptable for publication, you may indicate that here to bypass the “Comments to the Author” section, enter your conflict of interest statement in the “Confidential to Editor” section, and submit your "Accept" recommendation.

Reviewer #1: All comments have been addressed

2. Is the manuscript technically sound, and do the data support the conclusions?

Reviewer #1: Yes

3. Has the statistical analysis been performed appropriately and rigorously? 

Reviewer #1: Yes

4. Have the authors made all data underlying the findings in their manuscript fully available?

Reviewer #1: Yes

5. Is the manuscript presented in an intelligible fashion and written in standard English?

Reviewer #1: Yes

6. Review Comments to the Author

Reviewer #1: My comments have been addressed. The authors have responded to each of my questions and comments carefully.

7. PLOS authors have the option to publish the peer review history of their article (what does this mean?). If published, this will include your full peer review and any attached files.

Reviewer #1: No

---

## [Editor Report · Acceptance letter]

5 Dec 2023

PONE-D-23-20980R1 

Temporal dynamics of pro-inflammatory cytokines and serum corticosterone following acute sleep fragmentation in male mice 

Dear Dr. Ashley:

I'm pleased to inform you that your manuscript has been deemed suitable for publication in PLOS ONE. Congratulations! Your manuscript is now with our production department. 

Kind regards, 

on behalf of

Professor Dragan Hrncic 

Academic Editor

PLOS ONE